# The Effects of Spatial Characteristics and Visual and Smell Environments on the Soundscape of Waterfront Space in Mountainous Cities

Bingzhi Zhong [1], Hui Xie [2,3,*], Tian Gao [1], Ling Qiu [1], Heng Li [2] and Zhengkai Zhang [1]

1   College of Landscape Architecture and Art, Northwest A&F University, Yangling 712199, China
2   Faculty of Architecture and Urban Planning, Chongqing University, Chongqing 400044, China
3   Key Laboratory of New Technology for Construction of Cities in Mountain Area, Ministry of Education, Chongqing University, Chongqing 400044, China
*   Correspondence: xh@cqu.edu.cn

**Abstract:** The soundscape of waterfront space in mountainous cities (WSMC) can affect people's physical and mental health. Taking seven WSMCs in Chongqing, China, as the study area, this study aimed to investigate the soundscape and explore the influence of spatial characteristics and visual and smell environments on the soundscape of WSMCs through a sensewalking approach. The results show that the soundscape evaluations of WSMCs are of poor quality, and traffic sounds are dominant (33%). Among spatial characteristics, the position relative to the road (including vertical and horizontal distances) had a greater impact than other spatial indicators on soundscape evaluations. Elevation was positively correlated with the A-weighted equivalent sound level ($L_{Aeq}$) and negatively correlated with the soundscape comfort degree (SCD). In terms of visual elements, the proportions of paved ground, pedestrians, and buildings had negative effects on the soundscape, while those of the sky, water, and natural terrain had positive effects. High visual and smell environment quality can enhance soundscape evaluations, although the smell environment had a greater impact on the SCD than the visual environment in WSMCs. Finally, this study summarizes the recommended values of spatial characteristics and visual and smell environment indicators to put forward references for the soundscape design of WSMCs.

**Keywords:** soundscape; waterfront space in mountainous cities; spatial elements; multisensory interaction





## 1. Introduction

Urban waterfront space is a general term for a certain area connected by land and water in the city and is generally formed by water areas, water boundaries, and land areas [1]. Water spaces not only constitute a natural ecological transition between water and land to enrich urban landscapes but also foster a close connection between nature and people [2]. Waterfront space in mountainous cities (WSMC) has the characteristics of both urban waterfront space and mountainous topography, which creates landscape diversity and uniqueness in the residential environment [3]. With the spread of COVID-19, lockdowns and a decrease in outdoor activities led to an increase in both psychological stress and mortality by suicide [4]. There has been a growing demand for relaxation and entertainment by the public. As natural places, WSMCs can provide entertainment and perceived restoration to the public, with important health, ecological, and economic value.

Soundscapes have multiple impacts on environmental health. The World Health Organization (WHO) notes that high sound pressure levels (SPLs) can increase cardiovascular disease risk, sleep disturbance, and annoyance, which may reduce productivity and increase accident rates [5]. In contrast, studies have shown that there is a significant association between a positive soundscape and health-related effects, including increased

restoration and reduced stress-inducing mechanisms [6,7]. The soundscape is vital to building a distinctive cultural atmosphere, attracting visitors, increasing the district's vitality, and enhancing the landscape evaluation of an area [8]. Moreover, soundscape assessment can be used to monitor ecological conditions and reflect human disturbances to the ecosystem [9,10]. Some acoustic indices, such as acoustic complexity, acoustic evenness, and the normalized soundscape difference index (NDSI), are significantly correlated with biodiversity [11,12]. Therefore, the soundscape quality of WSMC can have important impacts on public well-being and environmental health. However, as most studies of WSMCs have focused on urban planning and landscape design based on visual features, the soundscape and the factors that influence soundscape evaluations have not yet been studied.

WSMCs have a variety of spatial elements that influence soundscape evaluations. As a natural spatial element, water bodies have special characteristics and important effects on the urban soundscape [13]. Watts et al. [14] found that the presence of water can create a sense of tranquility. The sound of water can not only mask noise but also increase the positive perception of urban green space [15]. However, urban waterfront space has different characteristics according to the different types and nature of adjacent water bodies. For example, in littoral areas, ocean visibility improves soundscape evaluation [16]. In areas close to streams or waterfalls, soundscape evaluation remains positive even with a high A-weighted equivalent sound level ($L_{Aeq}$) [17]. In addition, the soundscape of mountainous cities is closely related to the mountainous topography. Some studies have shown that, compared with flat ground, mountainous terrain may not only lead to an increase in the SPL but also limit people's activities, giving the soundscape specific spatial–temporal variations [18–20]. In addition, due to the dense urban roads and undulating terrain, the traffic noise in mountainous cities is often more complex than that in plains cities [21,22]. Scholars have achieved some understanding of the impact of single spatial characteristics. However, few studies have discussed the impact of various spatial elements on the soundscape with a background of multiple spatial characteristics, which is essential for WSMCs.

Multiple senses can convey more profound and comprehensive information than a single sense [23]. As important media for perceiving the environment, both vision and olfaction can interact with auditory perception [24,25]. Aural preferences and visual elements are intrinsically linked. The proportion of visual elements (such as buildings, vegetation, and sky), visual factors (such as distance and color), and visual perception indicators can all significantly affect soundscape evaluation [26–29]. Although studies on the interaction between auditory perception and olfaction have been limited to date, some studies have shown that odor can affect the response time to auditory stimuli, and the evaluation of sound and odor shows analogous trends of sensory comfort and preference [30,31]. Adams and Askins [32] found that sensewalking, as a varied method, can provide an effective way to study the urban environment from sensory perspectives, but existing studies on sensory interactions are mostly limited to the laboratory environment and have focused more on the influence of single sensory factors, resulting in limited external validity [33]. To date, few studies have considered both the perceptual characteristics and potential effects of visual and smell environments on the soundscapes of WSMCs in the field.

In order to bridge these gaps, this study investigated the soundscapes, spatial characteristics, and visual and smell environments of seven typical WSMCs in Chongqing, China, using the sensewalking approach. The aims of this study were divided into four parts. The first part was to investigate the current soundscape quality of WSMCs. The second part was to explore the influence of spatial characteristics on the soundscape of WSMCs. The third part was to explore visual environment–soundscape and smell environment–soundscape interactions. The fourth was to construct models to summarize the recommended variable values to achieve positive soundscape evaluations. Ensuring sufficient sampling sites and sensewalking participants, as well as organizing measurements under the influence of the epidemic, posed certain challenges in this study. Overall, the results of this study can provide data support and references for the soundscape design of WSMCs while providing

better-perceived restoration sites for the public and improving environmental health and people's happiness.

## 2. Materials and Methods

### 2.1. Studied Areas

Chongqing is a typical mountainous city in Southwest China, where 90% of the total area is mountains and hills. Located on both sides of the Yangtze and Jialing Rivers, seven WSMCs in Chongqing were randomly selected as the study areas, namely, Jiangbeizui (JB), Shacixiang (SC), Chaotianmen Square (CT), CBD Riverside Park (CB), Liziba Park (LZ), Jiulongpo Park (JL), and Nanbin Park (NB). Figure 1 shows their plans and the locations of walking points in Chongqing (JB: 6 points; SC: 9 points; CT: 9 points; CB: 10 points; LZ: 9 points; JL: 10 points; and NB: 10 points). The walking routes align with the main touring route in each WSMC. Table 1 provides basic information about the study areas.

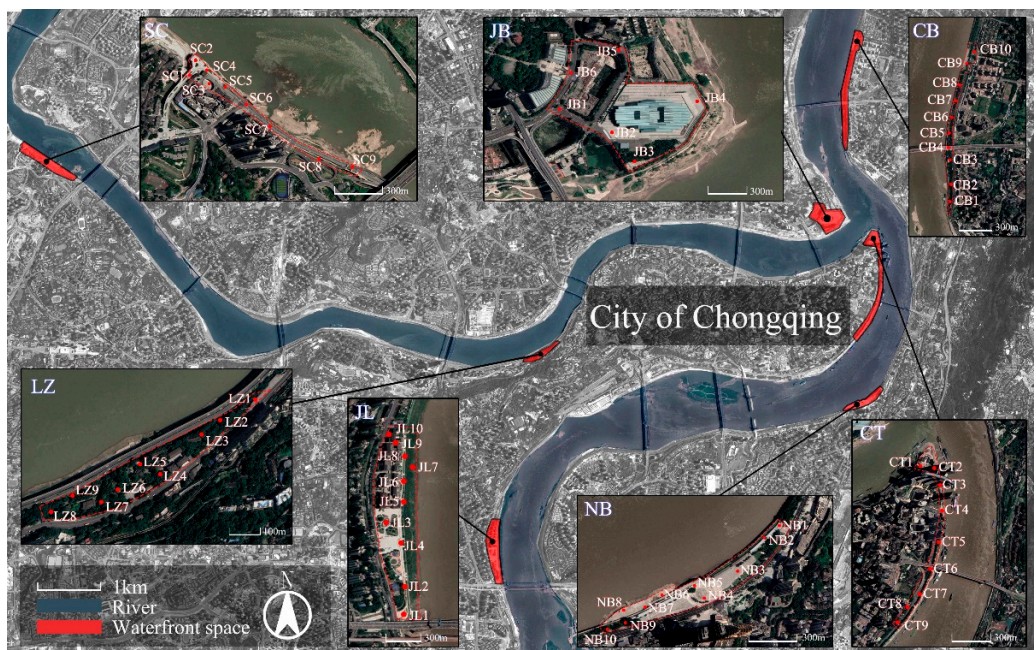

**Figure 1.** Plans of the seven studied WSMCs and the locations of walking points in Chongqing. WSMC = waterfront space in mountainous cities.

**Table 1.** Basic information on the seven WSMCs.

|  | JB | SC | CT | CB | LZ | JL | NB |
|---|---|---|---|---|---|---|---|
| Area (km$^2$) | 110.7 | 48.1 | 112.1 | 117.1 | 32.2 | 163.7 | 51.0 |
| Year of completion | 2009 | 2018 | 1998 | 2005 | 2010 | 2012 | 2005 |
| Affiliated urban district | Jiangbei | Shapingba | Yuzhong | Nan'an | Yuzhong | Jiulongpo | Nan'an |
| Function | Civic square | Business district | Business district | Park | Park | Civic square | Civic square |

### 2.2. The Sensewalking Approach and Questionnaire Design

Sensewalking is a common way to study one or more aspects of the sensory environment and usually involves a researcher walking alone or with one or more participants [34]. In this study, the subjective evaluations of the soundscape, as well as the visual and smell environments, were obtained using the sensewalking method. The participants comprised 172 architectural students (78 males and 94 females, with a mean age of 21 years old) from Chongqing University, with normal hearing and a basic knowledge of soundscapes and landscapes, who voluntarily participated in sensewalking. In total, 23–26 participants took part in sensewalking in each WSMC. They were chosen to understand future designers'

perspectives of the urban soundscape. Sensewalking was undertaken between 10 a.m. and 12 p.m. on summer weekdays. The weather conditions were stable, with a light breeze, no rain, and a temperature ranging from 25 to 32 °C (Figure 2). Each participant spent 5 min at each of the walking points to evaluate the soundscape quality and fill out the questionnaire. All participants underwent a pre-investigation in another local park before performing the formal sensewalk. The pre-investigation involved (a) familiarity with the survey process of sensewalking, (b) the rapid identification of sound sources and odors, and (c) the determination of appropriate subjective evaluation indicators of the soundscape and visual and smell environments.

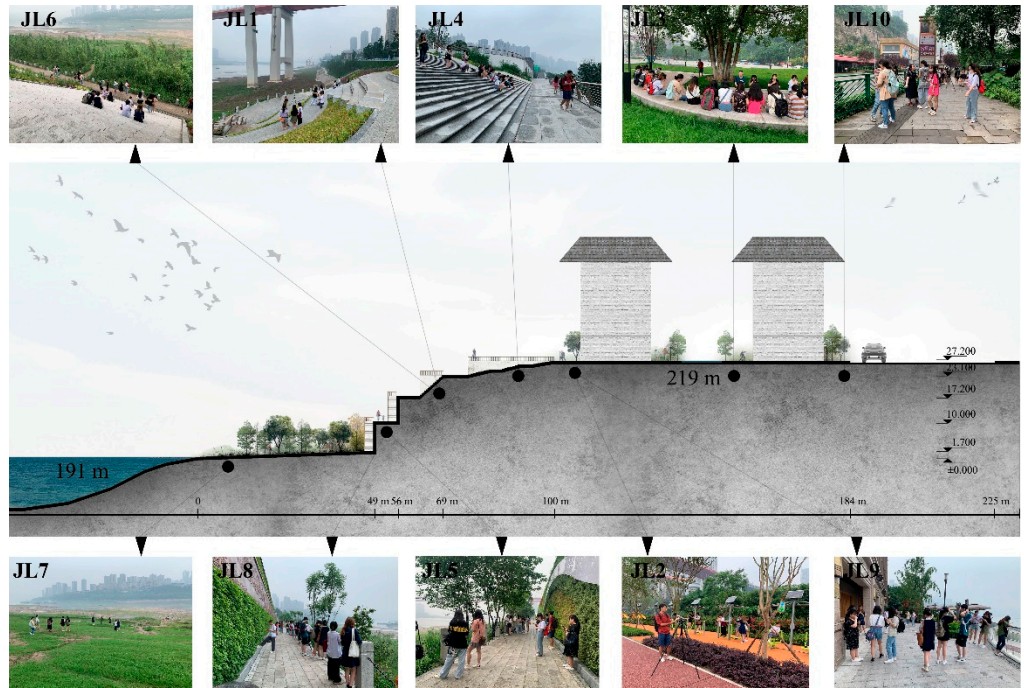

**Figure 2.** Sensewalking survey and mountainous topography status, taking JL as an example.

This study referred to the existing research and feedback on the pre-investigation design questionnaire. The questionnaire consists of three parts, covering the soundscape, visual environment, and smell environment. The subjective evaluation in this study was measured using a 5-point Likert scale.

In terms of the soundscape, in order to investigate the composition of the sound sources and the overall soundscape experience of the site, the questionnaire referred to the suggestions in ISO/TS 12913-2: 2018 and related research [35–37]. First, participants were asked to list all of the sound sources they noticed while listening at each walking point. The sound sources were classified as traffic sounds (e.g., cars, buses, trains, and airplanes), human sounds (e.g., conversation, laughter, children at play, and footsteps), natural sounds (e.g., biological and geophysical sounds), and mechanical sounds (e.g., sirens, construction, and industrial sounds). Second, the overall soundscape comfort was evaluated by the soundscape comfort degree (SCD), from 1 = "uncomfortable" to 5 = "comfortable".

In terms of the visual environment, previous studies have shown that comfort, complexity, and naturalness are valid for evaluating the visual landscape [38]. The subjective evaluation of the visual environment was obtained through the visual environment comfort degree (VECD), from 1 = "uncomfortable" to 5 = "comfortable"; the visual environment natural degree (VEND), from 1 = "artificial" to 5 = "natural"; and the visual environment diversity degree (VEDD), from 1 = "simple" to 5 = "complex".

In terms of the smell environment, few studies have explored criteria for subjective smell environment evaluations [39]. Therefore, this study used the smell environment comfort degree (SECD) as a subjective evaluation indicator to compare it with visual

and soundscape comfort (1 = "uncomfortable" to 5 = "comfortable"). In addition, the human sense of smell is currently the most sensitive tool available for assessing the smell environment [40]. Therefore, to identify the odor composition, participants were also asked to name the main odors at each walking point.

The details of the questionnaire are shown in Appendix A. Finally, 1544 valid questionnaires were obtained. With a KMO index of 0.806 and a Cronbach's Alpha of 0.791, the questionnaire had high validity and reliability.

### 2.3. Objective Soundscape Evaluation Parameter Measurement

In order to objectively evaluate the physical and ecological characteristics of the acoustic environments of the WSMCs, the $L_{Aeq}$ and NDSI were selected as the objective soundscape evaluation parameters in this study. The NDSI can be used to indicate human disturbances to biodiversity (such as the richness of bird species), ranging from $-1$ to 1 [10]. The higher the proportion of the artificial sound, the smaller the value. The formula is as follows, where $\alpha$ represents anthrophony (1–2 kHz), and $\beta$ represents biophony (2–11 kHz):

$$NDSI = \frac{(\beta - \alpha)}{(\beta + \alpha)}. \tag{1}$$

The $L_{Aeq}$ was calculated every 5 min ($L_{Aeq\_5min}$), and the audio (using a binaural method) was recorded at each walking point ($N = 63$) during the sensewalk. The $L_{Aeq\_5min}$ was measured using an AWA 6228$^+$ sound level meter (Class 1, Aihua Instruments Co., Ltd., Hangzhou, China). Audio samples were recorded using a PCM-M10 audio recorder (Sony Corporation, Tokyo, Japan). All equipment was located 1.2 m above the ground and more than 2 m from nearby buildings. ISO1996/2-2017 was followed throughout the field measurements [41]. RStudio (Version 1.1.463, RStudio, Inc., Boston, MA, USA) was used to analyze the NDSI from audio files. Detailed specifications of all measurement equipment and associated data processing software in this study are shown in Appendix B.

### 2.4. Spatial Indicator Measurements

Previous studies have shown that indicators of space, such as elevation, have significant impacts on soundscape evaluations in mountainous cities [42]. Based on the spatial characteristics of WSMCs, five indicators—elevation, vertical distance from the shoreline (VDS), horizontal distance from the shoreline (HDS), vertical distance from the road (VDR), and horizontal distance from the road (HDR)—were selected as spatial indicators in this study (Table 2) and measured using a YILI X28 altimeter (Appendix B).

**Table 2.** The ranges of spatial indicators of seven WSMCs. VDS = vertical distance from the shoreline. HDS = horizontal distance from the shoreline. VDR = vertical distance from the road. HDR = horizontal distance from the road.

|  | JB | SC | CT | CB | LZ | JL | NB |
|---|---|---|---|---|---|---|---|
| Elevation (m) | 244–276 | 232–247 | 271–278 | 189–210 | 196–229 | 192–218 | 200–213 |
| VDS (m) | 16.9–48.9 | 11.5–26.2 | 27.4–32.7 | 4.2–25.5 | 3.0–35.9 | 1.7–27.1 | 5.4–19.2 |
| HDS (m) | 109.0–410.2 | 9.0–98.5 | 65.7–74.7 | 9.4–84.8 | 30.6–104.0 | 29.5–151.7 | 18.7–80.6 |
| VDR (m) | 6.4–238.9 | 9.6–187.7 | 8.1–116.8 | 29.9–74.3 | 15.1–46.2 | 35.2–184.9 | 17.3–76.1 |
| HDR (m) | −25.6–6.4 | 9.2–5.5 | −3.7–5.5 | −21.5−−0.2 | −32.6–0.3 | −22.7–2.7 | −11–2.8 |

### 2.5. Identification of the Proportion of Visual Elements

In addition, to obtain the visual elements experienced by participants during the sensewalk, panoramic street-view images were taken using smartphones (Appendix B) at each walking point. A fully connected network (FCN) model (GUC. HPSCIL, University of Geo-sciences, China) was used to identify different visual elements in street-view images (Figure 3) [43]. Coupled with the calculation of the per-pixel loss, the FCN produces the area ratio of each visual element in the image by counting the number of pixels in each

segmentation mask. The FCN model can extract 150 visual elements and achieve a pixel-wise accuracy of 81% for training data and 67% for actual data (Appendix B). Seven visual elements that occupy a relatively large proportion of WSMCs were identified in this study, namely, paved ground, buildings, vegetation, sky, water, natural terrain, and pedestrians and animals.

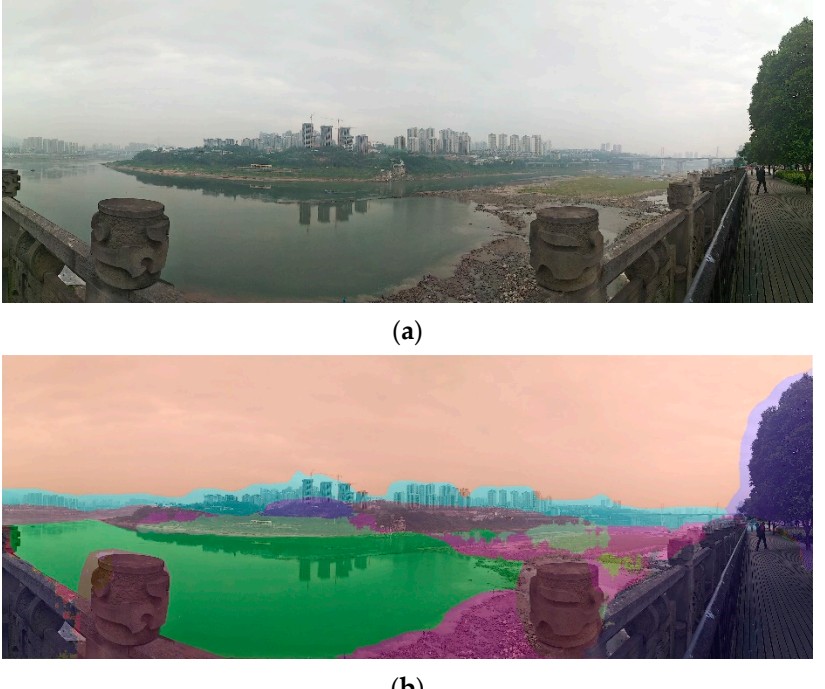

(**a**)

(**b**)

**Figure 3.** Visual element recognition in street-view images: (**a**) raw image and (**b**) the same image segmented with the FCN model (fully connected network).

*2.6. Data Analysis*

Based on the one-sample Kolmogorov–Smirnov test, data were not normally distributed. Spearman's rho correlation analysis was performed to identify the relationships between spatial indicators (elevation, VDS, HDS, VDR, and HDR), the proportions of visual elements, subjective evaluations of the visual and smell environments (VECD, VEND, VEDD, and SECD), and soundscape evaluation parameters ($L_{Aeq\_5min}$, NDSI, and SCD). The Spearman's correlation coefficients r and *p* were used to find correlations between variables and soundscape evaluation parameters; a *p*-value of less than 0.05 was considered statistically significant.

Previous studies have shown that some important variables for soundscape evaluations, such as distance and environment subjective evaluations, behave linearly [44,45]. The final aim of this study was to summarize the recommended values of the variables and provide references for the design of WSMCs. Therefore, in order to further explore the relationship between variables and soundscape evaluation parameters, multiple linear regression analyses were used to model spatial indicators, the proportions of visual elements, and subjective evaluations of visual and smell environments (dependent variables) with soundscape evaluation parameters (independent variables). Adjusted $R^2$ and β coefficients were used to assess the quality of the obtained models. Variables with $p < 0.05$ and VIF < 2 were retained in the model using the stepwise method.

All statistical analyses were performed using Statistical Package for Social Scientists (SPSS) software version 22.0 (IBM SPSS, Inc., Chicago, IL, USA). The minimum hardware required for data analysis was a personal computer with a 2.5 GHz Intel Core i5 processor and at least 4 GB of RAM.

## 3. Results

### 3.1. Soundscape Evaluations of Waterfront Spaces in Mountainous Cities (WSMCs)

The sound sources listed by participants were classified as traffic sounds, human sounds, natural sounds, and mechanical sounds according to ISO/TS 12913-2: 2018 [35], and the proportions of sound sources in seven studied WSMCs were calculated (Table 3). It is known that the high $L_{Aeq}$ of traffic noise can affect soundscape comfort, while natural sounds can significantly improve soundscape comfort [46]. However, the results showed that, on average, the proportion of traffic sounds was 33%, higher than in urban parks in mountainous cities (from 4.9% to 9.2%), while the proportion of natural sounds was only 27%, lower than in urban parks in mountainous cities (from 31.4% to 53.3%) [19]. This indicates that the soundscape components of WCMCs need to be improved by controlling the interference of traffic noise and improving the proportion of natural sounds.

**Table 3.** Proportions of traffic sounds, human sounds, natural sounds, and mechanical sounds in seven WSMCs.

|  | JB | SC | CT | CB | LZ | JL | NB | Mean |
|---|---|---|---|---|---|---|---|---|
| Traffic sounds | 34% | 20% | 42% | 36% | 37% | 27% | 37% | 33% |
| Human sounds | 23% | 21% | 28% | 15% | 20% | 27% | 18% | 22% |
| Natural sounds | 25% | 28% | 14% | 37% | 28% | 32% | 23% | 27% |
| Mechanical sounds | 18% | 32% | 16% | 11% | 15% | 14% | 22% | 18% |

Figure 4 shows the mean values of the soundscape evaluation parameters ($L_{Aeq\_5min}$, NDSI, and SCD) at walking points in seven WSMCs. It can be seen that the overall $L_{Aeq\text{-}5min}$ of WSMCs was high (Figure 4a). In fact, the $L_{Aeq\text{-}5min}$ values of the six WSMCs and 79% of the walking points were higher than 55 dBA, exceeding the national recommended value (daytime, city park, and green space) [47]. All NDSIs of WSMCs were negative, ranging from −0.425 to −0.004 (Figure 4b). This indicates that human disturbance was dominant in the WSMCs. The NDSI was the highest (−0.004) in JL and the lowest (−0.425) in NB. This might be related to the fact that JL is located in the old town and the pedestrian flow was low, while NB is located in a popular scenic spot and the pedestrian flow was high. The SCD of the seven WSMCs was in the range of 2.4–3.0, between "a little uncomfortable" and "moderate" (Figure 4c). Specifically, only 31.7% of the walking points were positively evaluated in terms of the SCD. In general, the soundscape evaluations of WSMCs were of poor quality.

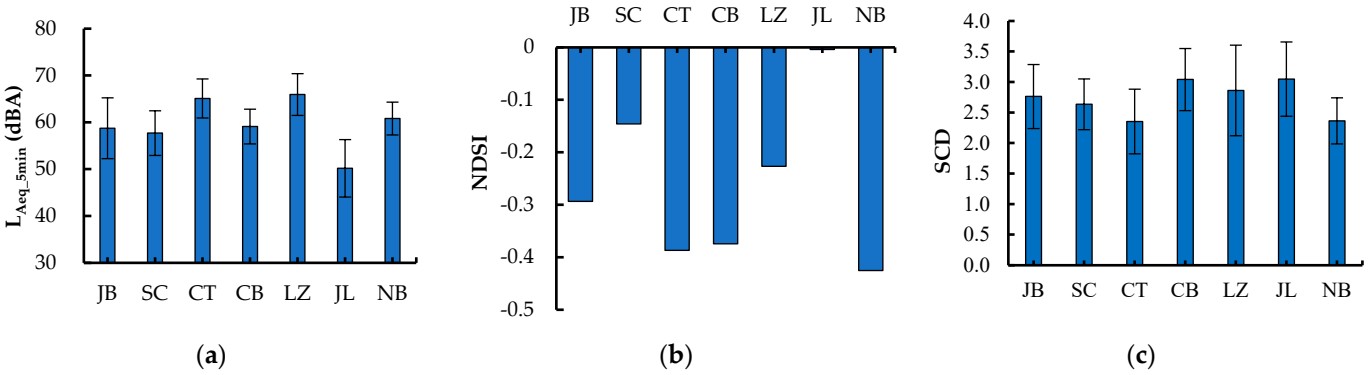

**Figure 4.** The mean values of soundscape evaluation parameters at walking points in seven WSMCs. (**a**) $L_{Aeq\_5min}$, (**b**) NDSI, and (**c**) SCD. $L_{Aeq\_5min}$ = A-weighted equivalent sound level calculated every 5 min; NDSI = normalized soundscape difference index; SCD = soundscape comfort degree. All soundscape evaluation parameters are the average values for the area.

*3.2. The Influence of Spatial Characteristics on the Soundscape*

Combined with the spatial characteristics of WSMCs, spatial indicators, including the elevation, VDS, HDS, VDR, and HDR, were extracted for correlation analysis using soundscape evaluation parameters (Table 4). The results show that both elevation and VDS had significantly positive correlations with the $L_{Aeq\_5min}$ (r = 0.323 and 0.344, respectively, $p < 0.01$) and significantly negative correlations with the SCD (r = −0.375 and −0.344, respectively, $p < 0.01$). This was due to the significant autocorrelation between elevation and VDS (r = 0.734, $p < 0.01$). In terms of road-related indicators, the VDR had a significantly negative correlation with the SCD (r = −0.450, $p < 0.01$). The HDR had a significantly negative correlation with the $L_{Aeq\_5min}$ (r = −0.635, $p < 0.01$) and significantly positive correlations with the NDSI and SCD (r = 0.306 and 0.402, respectively, $p < 0.01$). This indicates that, regardless of whether in the vertical or horizontal direction, the closer the distance to the road, the lower the comfort of the soundscape. The correlation coefficient is a statistical tool used to measure the extent of the relationship between variables. Since the HDR and VDR are the spatial indicators with the highest correlation coefficients with the $L_{Aeq\_5min}$ (r = −0.635) and SCD (r = −0.450), respectively, the HDR may have a greater influence on the $L_{Aeq\_5min}$, and the VDR may have a greater influence on the SCD.

**Table 4.** Spearman's rho correlation coefficients for the relationship between spatial indicators and soundscape evaluation parameters.

|  | Elevation | VDS | HDS | VDR | HDR |
|---|---|---|---|---|---|
| $L_{Aeq\_5min}$ | 0.323 ** | 0.344 ** | 0.049 | 0.112 | −0.635 ** |
| NDSI | −0.103 | −0.143 | −0.079 | −0.145 | 0.306 ** |
| SCD | −0.375 ** | −0.344 ** | −0.087 | −0.450 ** | 0.402 ** |

Notes: ** significance at the 0.01 level (2-tailed).

In contrast to Spearman's correlation coefficients, multiple linear regression found that the VDS was left out in the $L_{Aeq\_5min}$ model, and elevation and VDS were left out in the SCD model (Table 5). A possible reason for this might be the multicollinearity of the VDS with the $L_{Aeq\_5min}$ (VIF = 2.875), and the correlations of elevation and the VDS with the SCD were relatively low. Furthermore, the spatial indicators accounted for 42.5%, 21.7%, and 35.7% of the variability in the $L_{Aeq\_5min}$, NDSI, and SCD, respectively (adjusted $R^2$ = 0.425, 0.217, and 0.357, respectively). Referring to similar soundscape studies, an adjusted $R^2$ over 0.3 provides sufficient reliability for the linear model [27,48]. It can be seen that the performance of the linear regression model of the NDSI was relatively low. This may mean that although the linear regression model was significant, the impact of the HDR on the NDSI was limited. The most influential variables on the $L_{Aeq\_5min}$, NDSI, and SCD were the HDR (β = −0.578), HDR (β = 0.479), and VDR (β = 0.503), respectively.

**Table 5.** Results of multiple linear regression analyses of spatial indicators and soundscape evaluation parameters.

| Dependent Variables | Factors | Adjusted $R^2$ | β | SE | *t*-Value | *p* | VIF |
|---|---|---|---|---|---|---|---|
| $L_{Aeq\_5min}$ | HDR | 0.425 | −0.578 | 0.011 | −5.967 | 0.000 | 1.013 |
| | Elevation | | −0.271 | 0.023 | 2.791 | 0.007 | 1.013 |
| NDSI | HDR | 0.217 | 0.479 | 0.001 | 4.264 | 0.000 | 1.000 |
| SCD | VDR | 0.357 | −0.503 | 0.008 | −4.727 | 0.000 | 1.094 |
| | HDR | | 0.236 | 0.001 | 2.213 | 0.031 | 1.094 |

Linear regression curves between spatial indicators and soundscape evaluation parameters are shown in Figure 5. It can be seen that when the HDR was above 90 m, the $L_{Aeq\_5min}$ could drop below 55 dBA; when the elevation was below approximately 220 m, the $L_{Aeq\_5min}$ of WSMCs could drop below 55 dBA (Figure 5a). In terms of the NDSI, when

the HDR was more than 90 m, the NDSI became positive (Figure 5b). This indicates that, at this distance, there was a higher proportion of biological sounds, and a better ecological status could be achieved. In terms of the SCD, this study found that when the HDR was more than 70 m, or the VDR was less than $-10$ m, the evaluation of the SCD was higher than moderate (evaluation = 3). Small changes in the vertical distance could result in large variances in the SCD (Figure 5c).

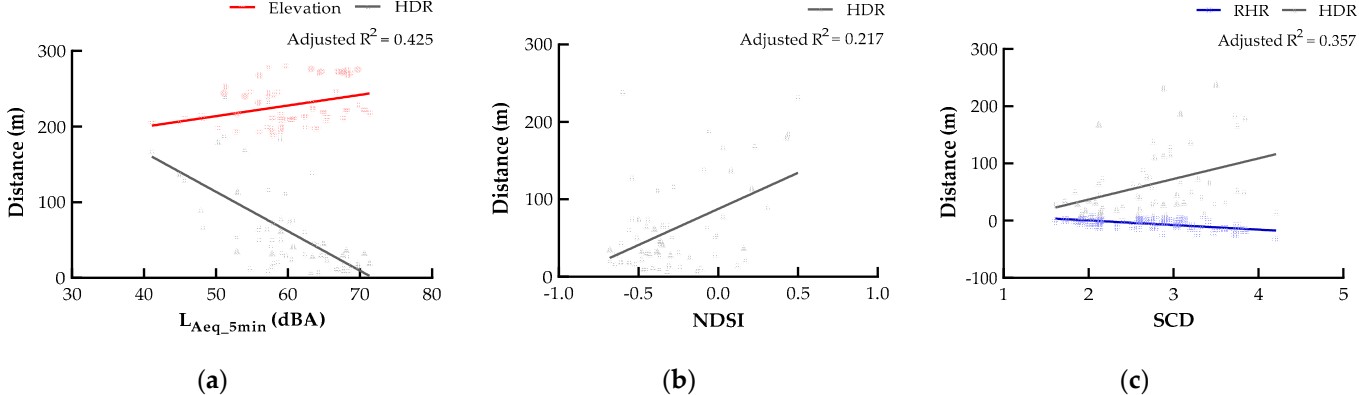

**Figure 5.** Linear regression curves between spatial indicators and soundscape evaluation parameters: (**a**) spatial indicators and $L_{Aeq\_5min}$; (**b**) spatial indicators and NDSI; (**c**) spatial indicators and SCD.

### 3.3. The Influence of Visual and Smell Environments on the Soundscape

#### 3.3.1. Visual Environment of WSMCs

Table 6 shows the average proportions of visual elements in seven WSMCs. Paved ground and buildings can form an artificially closed spatial enclosure. It was found that the average proportion of paved ground and buildings were 14.6% and 20.8%, respectively, and their total proportion reached 41%. The average proportion of vegetation was 20.8%. In terms of the sky, except for the proportion of sky in LZ, which was only 4.1%, little difference was found between the other six WSMCs, which were within the range of 10.0%–25.4%. LZ is located in a historic district where the trees are leafy, covering the sky. The waterfront is an important feature of WSMCs. However, water only accounted for 0.1%–7.2% of the seven WSMCs, with an average of only 2.6%. In fact, most of the waterfront spaces in the studied WSMCs were designed to be open for viewing water. The reason for the small proportion of water was that there were few waterfront spaces in the planning of the WSMCs, and many walking points were sheltered by vegetation and buildings. Other visual elements, such as natural terrain and pedestrians and animals, made up a relatively small proportion of WSMCs and were within the range of 0.2%–1.9%.

**Table 6.** Average proportions of nine visual elements in seven WSMCs.

|  | JB | SC | CT | CB | LZ | JL | NB | Average |
|---|---|---|---|---|---|---|---|---|
| Paved ground | 33.0% | 30.7% | 26.9% | 21.1% | 32.7% | 16.5% | 24.1% | 26.4% |
| Buildings | 20.1% | 17% | 23.7% | 10.7% | 8.2% | 10.1% | 12.1% | 14.6% |
| Vegetation | 16.8% | 8.8% | 19.9% | 24.9% | 26.7% | 28.1% | 20.4% | 20.8% |
| Sky | 15.8% | 25.4% | 10% | 14.5% | 4.1% | 17.7% | 20.4% | 15.4% |
| Water | 0.1% | 4.5% | 1.6% | 7.2% | 1.3% | 1.0% | 2.3% | 2.6% |
| Natural terrain | 0.4% | 1.3% | 0.9% | 4.5% | 0.7% | 0.5% | 0.7% | 1.3% |
| Pedestrians and animals | 2.1% | 0.5% | 1.4% | 0.1% | 8.6% | 0.2% | 0.6% | 1.9% |

In terms of the subjective evaluation of the visual environment, the results of the VECD, VEND, and VEDD are shown in Figure 6. Five WSMCs' results for the VECD were higher than 3 (moderate), indicating that the overall visual environment of WSMCs was relatively comfortable. However, the VEND and VEDD were generally lower. Additionally, the

results of the VEND were lower than those of VEDD at 2.5–3.2, between "a little artificial" and "a little natural". This indicates that the visual environments of WSMCs were still lacking richness in landscape diversity, especially the natural landscape.

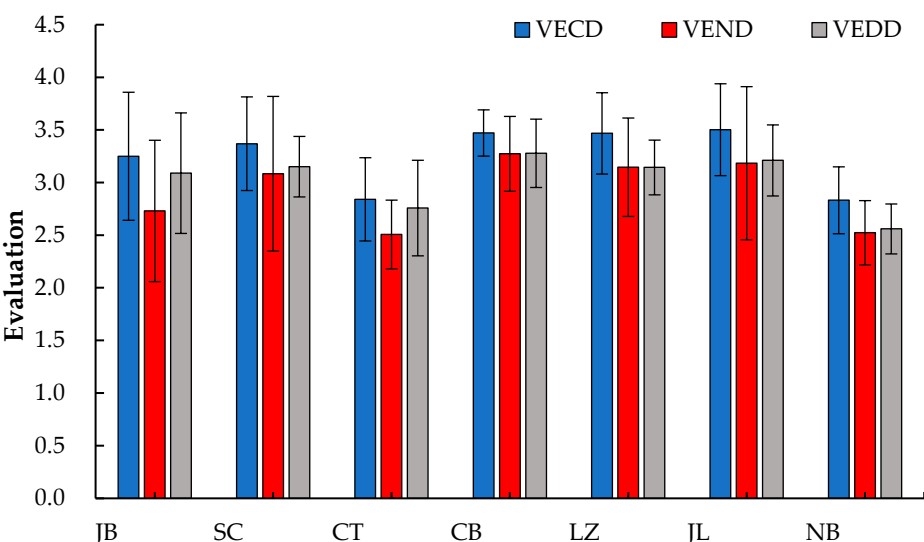

**Figure 6.** Results of VECD, VEND, and VEDD in seven WSMCs. VECD = visual environment comfort degree; VEND = visual environment natural degree; VEDD = visual environment diversity degree.

3.3.2. Smell Environment of WSMCs

According to common odor sources in the city [49], the odors identified by the participants during the sensewalk were classified into five categories: natural odors, emission odors, food odors, building material odors, and human odors (Table 7). It can be seen that, in the WSMCs, the proportion of natural odors was the highest, with an average of 66.2%. The second highest proportion was emission odors, ranging from 7.5% to 32.3%. The proportions of food odors, building material odors, and human odors were lower, no more than 6%. In terms of the subjective evaluation of the smell environment, the results of the SECD in seven WSMCs are shown in Figure 7. It can be seen that the SECDs were higher with a large proportion of natural odors (such as CB and JL) and lower with a large proportion of emission odors (e.g., CT). In general, the scores of the SECD ranged from 2.7 to 3.3, and the five WSMCs' SECD evaluation results were higher than 3 (moderate).

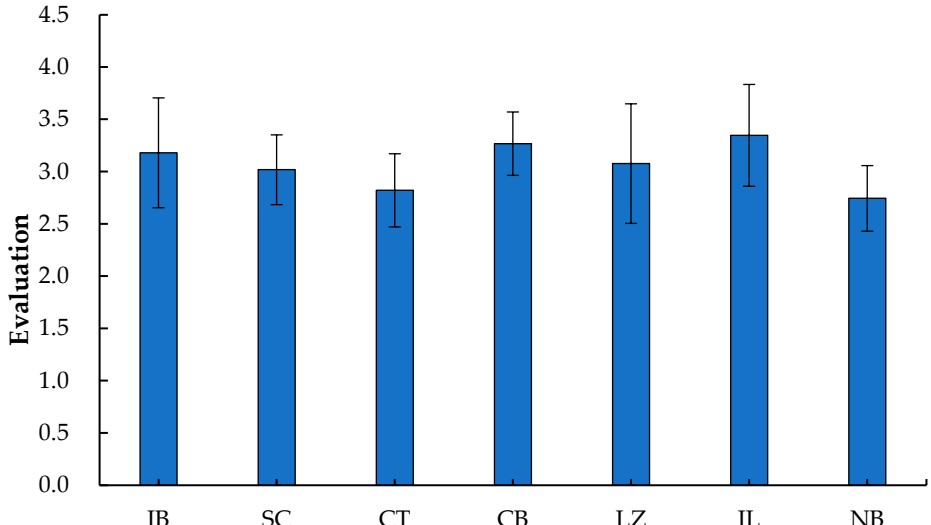

**Figure 7.** Evaluation results of SECD in seven WSMCs. SECD = smell environment comfort degree.

**Table 7.** Average proportions of five categories of odors in seven WSMCs.

|  | JB | SC | CT | CB | LZ | JL | NB | Average |
|---|---|---|---|---|---|---|---|---|
| Natural odors | 63.2% | 56.5% | 43.3% | 82.5% | 70.1% | 81.1% | 66.7% | 66.2% |
| Emission odors | 25.3% | 17.6% | 32.3% | 7.6% | 20.4% | 7.5% | 18.5% | 18.5% |
| Human odors | 3.4% | 5.1% | 7.0% | 9.2% | 4.4% | 0.0% | 11.1% | 5.7% |
| Building material odors | 6.9% | 7.7% | 3.5% | 0.7% | 5.3% | 11.3% | 3.7% | 5.6% |
| Food odors | 1.1% | 11.9% | 13.9% | 0.0% | 0.0% | 0.0% | 0.0% | 3.8% |

### 3.3.3. The Influence of Visual and Smell Environments on Soundscape Evaluation Parameters

The correlation analyses between the proportions of visual elements and soundscape evaluation parameters are shown in Table 8. It was found that the $L_{Aeq\_5min}$ was significantly positively correlated with the proportions of paved ground and pedestrians and animals (r = 0.276, $p < 0.05$; r = 0.632, $p < 0.01$, respectively) and significantly negatively correlated with the proportions of sky and water (r = −0.460, $p < 0.01$; r = −0.255, $p < 0.05$, respectively). In terms of the SCD, it was found that the proportion of buildings was significantly negatively correlated with the SCD (r = −0.254, $p < 0.05$), and the proportions of water and natural terrain were significantly positively correlated with the SCD (r = 0.262 and 0.311, $p < 0.05$, respectively). This indicates that people prefer the soundscape experience of waterfront space and natural terrain space (such as a natural terrace, slope, or valley), rather than a building-dominated soundscape experience. In general, the results show that the proportions of paved ground, buildings, and pedestrians and animals have negative effects on the soundscape, while the sky, water, and natural terrain have positive effects. In terms of correlation coefficients, pedestrians and animals and natural terrain have the greatest impacts on the $L_{Aeq\_5min}$ (r = 0.632) and SCD (r = 0.311), respectively.

**Table 8.** Spearman's rho correlation coefficients for the relationship between seven visual elements and soundscape evaluation parameters.

|  | Paved Ground | Buildings | Vegetation | Sky | Water | Natural Terrain | Pedestrians and Animals |
|---|---|---|---|---|---|---|---|
| $L_{Aeq\_5min}$ | 0.276 * | 0.026 | 0.214 | −0.460 ** | −0.255 * | −0.147 | 0.632 ** |
| NDSI | −0.010 | −0.013 | −0.139 | 0.139 | 0.079 | 0.086 | −0.214 |
| SCD | −0.181 | −0.254 * | −0.003 | 0.108 | 0.262 * | 0.311 * | 0.185 |

Notes: * significance at the 0.05 level (2-tailed); ** significance at the 0.01 level (2-tailed).

All subjective evaluations of visual and smell environments were negatively correlated with the $L_{Aeq\_5min}$ and positively correlated with the SCD, and all of the visual environment subjective evaluations were positively correlated with the NDSI ($p < 0.01$), as demonstrated in Table 9. This indicates that visual and smell environments can enhance the soundscape evaluation, which confirms the association between visual and olfactory perceptions in soundscape evaluations [30,50]. Only the SECD was not significantly correlated with the NDSI ($p > 0.05$). This might indicate that the smell environment, as perceived by humans, has little effect on the ecological characteristics of the acoustic environment. In terms of the SCD, it is worth noting that the SCD was more strongly correlated with the SECD (r = 0.780) than the VECD (r = 0.729). This indicates that the smell environment had a greater impact on the SCD than the visual environment in WSMCs.

From the multiple linear regression analyses (Table 10), it can be seen that subjective evaluations of visual and smell environments accounted for 24.9%, 12.7%, and 69.6% of the variability in the $L_{Aeq\_5min}$, NDSI, and SCD (adjusted $R^2$ = 0.249, 0.127, and 0.696, respectively). The adjusted $R^2$ values of subjective evaluations for the $L_{Aeq\_5min}$ and NDSI were lower than 0.3. The impact of the VECD and VEDD on the $L_{Aeq\_5min}$ and NDSI was limited. In a study on the relationship between visual elements and the soundscape, Liu, Kang, Behm and Luo [26] found that the adjusted $R^2$ of the proportions of visual elements

was rather low (adjusted $R^2 > 0.1$). The proportions of visual elements account for 21.7% and 4.9% of the variability in the $L_{Aeq\_5min}$ and SCD, respectively (adjusted $R^2 = 0.217$ and 0.049, respectively). The impact of buildings on the SCD was limited. In addition, pedestrians and animals were the most influential variable on the $L_{Aeq\_5min}$ ($\beta = 0.376$), and buildings was the most influential variable on the SCD ($\beta = -0.254$). In the subjective evaluations of visual and smell environments, the SECD was the most influential variable on the SCD ($\beta = 0.553$). The VECD and VEDD were the most influential variables on the $L_{Aeq\_5min}$ and NDSI, respectively ($\beta = -0.511$ and 0.376, respectively).

**Table 9.** Spearman's rho correlation coefficients for the relationships between subjective evaluations of visual and smell environments and soundscape evaluation parameters.

| | VECD | VEND | VEDD | SECD |
|---|---|---|---|---|
| $L_{Aeq\_5min}$ | −0.525 ** | −0.482 ** | −0.400 ** | −0.506 ** |
| NDSI | 0.328 ** | 0.305 ** | 0.327 ** | 0.195 |
| SCD | 0.729 ** | 0.708 ** | 0.566 ** | 0.780 ** |

Notes: ** significance at the 0.01 level (2-tailed).

**Table 10.** Results of multiple linear regression analyses for the proportions of visual elements and subjective evaluations of visual and smell environments with soundscape evaluation parameters.

| Dimensions | Dependent Variables | Factors | Adjusted $R^2$ | β | SE | *t*-Value | *p* | VIF |
|---|---|---|---|---|---|---|---|---|
| The proportions of visual elements | $L_{Aeq\_5min}$ | Pedestrians and animals | 0.217 | 0.376 | 22.081 | 3.325 | 0.002 | 1.013 |
| | | Paved ground | | 0.278 | 5.272 | 2.453 | 0.017 | 1.013 |
| | SCD | Buildings | 0.049 | −0.254 | 0.762 | −2.055 | 0.044 | 1.000 |
| Subjective evaluations of visual and smell environments | $L_{Aeq\_5min}$ | VECD | 0.249 | −0.511 | 1.505 | −4.647 | 0.000 | 1.000 |
| | NDSI | VEDD | 0.127 | 0.376 | 0.080 | 3.170 | 0.002 | 1.000 |
| | SCD | SECD | 0.696 | 0.553 | 0.122 | 5.920 | 0.000 | 1.778 |
| | | VEND | | 0.365 | 0.093 | 3.906 | 0.000 | 1.778 |

The linear regression curves between the proportions of visual elements and soundscape evaluation parameters are shown in Figure 8a,b. It can be seen that when the proportion of paved ground was lower than 22%, or the proportion of pedestrians and animals was lower than 1%, the $L_{Aeq\_5min}$ could drop below 55 dBA (Figure 8a); when the proportion of buildings was less than 13%, the SCD could reach "moderate" (evaluation score = 3) or above (Figure 8b). In linear regression curves between subjective evaluations of visual and smell environments and soundscape evaluation parameters, when the VECD reached 3.4 or above, the $L_{Aeq\_5min}$ was below 55 dBA (Figure 8c); when the VEDD was above 3.2, the NDSI could reach a positive value (Figure 8d). It should be noted that, in terms of the SCD, only when the VEND and SECD reached 3.1 and 3.2, respectively, could the evaluation of the SCD reach "moderate" (evaluation score = 3) or above (Figure 8e). This indicates that the improvement in soundscape comfort in WSMCs might require a better natural visual environment and a comfortable smell environment.

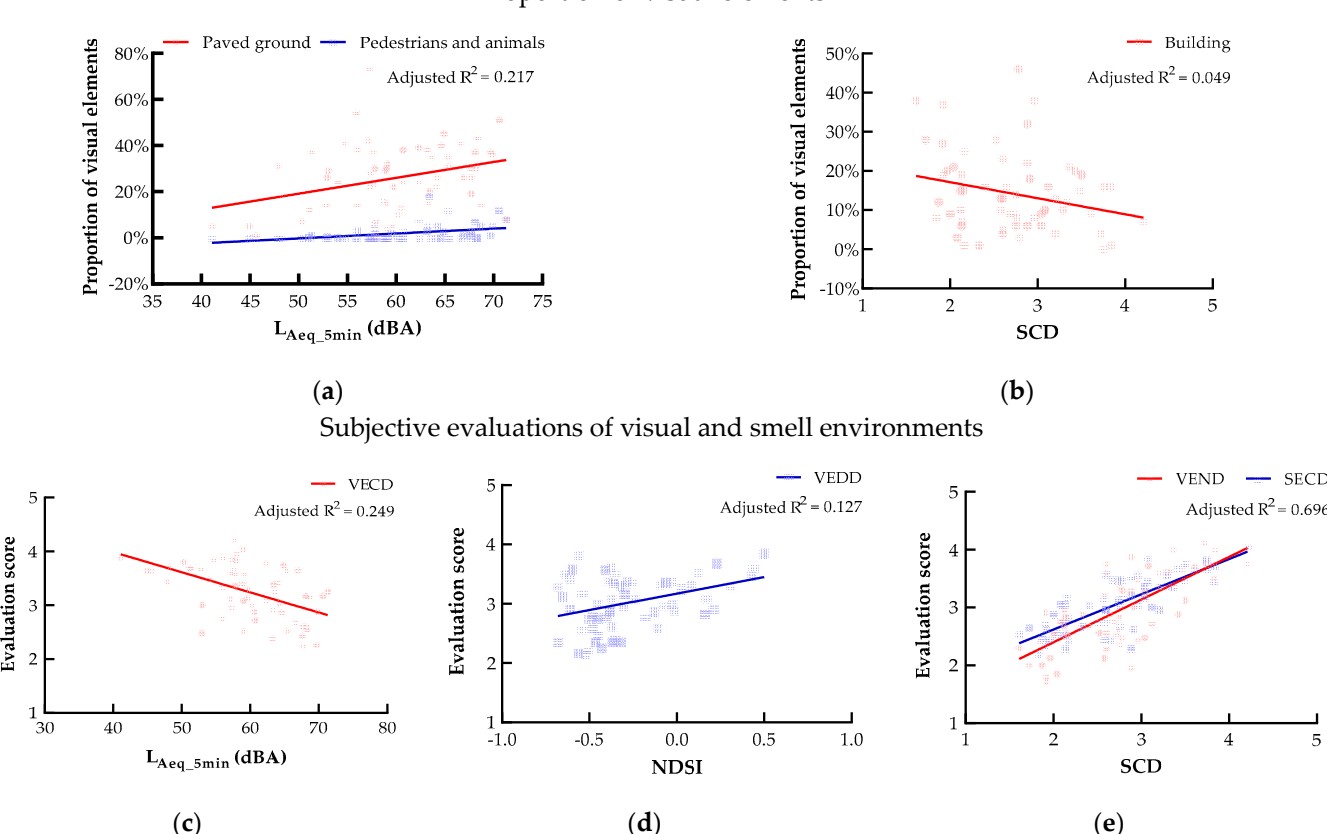

**Figure 8.** Linear regression curves between the proportion of visual elements and soundscape evaluation parameters, and subjective evaluations of visual and smell environments and soundscape evaluation parameters: (**a**) the proportion of visual elements and $L_{Aeq\_5min}$; (**b**) the proportion of visual elements and SCD; (**c**) subjective evaluations of visual and smell environments and $L_{Aeq\_5min}$; (**d**) subjective evaluations of visual and smell environments and NDSI; (**e**) subjective evaluations of visual and smell environments and SCD.

## 4. Discussion

### 4.1. The Influence of Spatial Characteristics on the Soundscape

This study found that the soundscape evaluations of WSMCs could be affected by spatial characteristics, such as elevation, VDS, VDR, and HDR. Due to excessive noise from urban expressways caused by the terrain fluctuation and compact urban structure in mountainous cities [22], regression analyses showed that the position relative to the road (including VDR and HDR) had a greater impact than other spatial indicators on the WSMC soundscape parameters. In a field study of a mountain landscape, Liu, Kang and Meng [42] found that elevation was significantly negatively correlated with sharpness but had no significant effects on the SPL or SCD. In contrast, this study found that elevation was significantly correlated with the $L_{Aeq\_5min}$ and SCD in WSMCs. This may be due to the fact that WSMCs are greatly influenced by urban elements, such as roads and pedestrians. In addition, spatial variations in soundscapes may lead to the distribution of biodiversity along elevation gradients at large spatial scales [51]. This study found that the NDSI was only significantly correlated with the VDR and HDR, but not with elevation, the VDS, or the HDS. This indicates that topography has little impact on biodiversity in small-scale spaces, such as WSMCs, and more attention should be paid to the impact of the spatial distribution of noise sources, such as traffic noise, on the ecological environments of WSMCs. Laboratory studies have shown that a horizontal position near water can significantly improve the evaluation of soundscape comfort by using photographs [52]. However, this study found that, in WSMCs, only the vertical position near water was related to the SCD. Additionally,

the correlation coefficient between the VDR and SCD was the highest (r = −0.572). This might suggest that vertical spatial indicators have a greater impact on soundscape comfort in WSMCs.

### 4.2. The Influence of Visual and Smell Environments on Soundscape

In terms of visual elements, this study found that paved ground, buildings, sky, water, natural terrain, and pedestrians and animals were effective landscape elements influencing the soundscape evaluation parameters. Unlike the sound of fountains, this study found that water sounds from rivers have little effect on the SPL in WSMCs [53]. The presence of water can still cause a sense of visual tranquility and improve the comfort of the soundscape [14], but the visual element of the water accounted for only 0.1%–7.2% in the studied WSMCs. This indicates that it is necessary to increase the number of walking points where water can be seen, or build other waterscapes to improve the visibility of water and the proportion of water sounds in WSMCs. In terms of subjective evaluation, the regression analysis found that the diversity of the visual environment is an important factor affecting the proportion of biological sounds. However, this study also found that the visual landscape of WSMCs is still not rich in landscape diversity.

In terms of the smell environment, although the overall evaluation of the SECD tended to be positive, it should be noted that emission odors (such as traffic and waste emissions) accounted for large proportions, ranging from 7.5% to 32.3%. A previous study found that there was a strong similarity between the soundscape and the smell environment evaluation [30]. However, this study found that the SECD was not significantly correlated with the NDSI. This may be due to the fact that biologically emitted odors are less detectable than biological visual elements. Notably, in the regression model, the SECD was found to be the most influential variable on the SCD (β = 0.553). This may be because the olfactory experience has a greater impact on subjective feelings (including subjective emotions and environmental and spatial memory) than visual perception [31,54].

### 4.3. Suggestions for Soundscape Improvement in WSMCs

Specifically, this study summarizes the recommended values of specific spatial indicators, the proportions of visual elements, and subjective evaluations of visual and smell environments through linear regression curves so as to achieve a positive soundscape evaluation (Table 11). These data can provide suggestions and references for healthy acoustic environment design and the study of WSMCs in the future. It is worth noting that the adjusted $R^2$ values of the regression models of the $L_{Aeq}$ and NDSI using spatial indicators (0.425 and 0.217, respectively) were higher than those using subjective evaluations of visual and smell environments (0.249 and 0.127, respectively). In contrast, the adjusted $R^2$ value of the regression model of the SCD using subjective evaluations of visual and smell environments (0.696) was higher than that of the regression model using spatial indicators (0.249). This indicates that the objective evaluation of the soundscape is more affected by spatial indicators, and soundscape comfort is more affected by visual and smell environments.

In addition, this study has some limitations that need to be addressed in the future. First, although the linear regression model was effective, it was found that the performance of the models of the VECD and $L_{Aeq\_5min}$, the HDR and VEDD and the NDSI, and buildings and the SCD was relatively low (Table 11). The limitation of photography technology may lead to a low adjusted $R^2$ in the visual element model [26]. In addition, the removal of strong difference points may lead to a higher adjusted $R^2$ coefficient (such as the Grubbs test) [55]. Future studies should discuss how to improve the adjusted $R^2$ coefficient and obtain more accurate recommended values. Secondly, the participants in this study were architecture students. However, participants with different social backgrounds may influence the results of environmental perception [56]. In follow-up studies, randomized participants might be employed across multiple areas to verify the generality of the conclusions of this study.

**Table 11.** The recommended values to achieve positive soundscape evaluation in terms of spatial characteristics and visual and smell environments.

| Soundscape Evaluation Parameters | Objectives | Indicator of Spatial Characteristics, Visual and Smell Environments | | Recommended Values |
|---|---|---|---|---|
| $L_{Aeq\_5min}$ | ≤55dBA | Spatial indicators | HDR | ≥90 m |
| | | | Elevation | ≥220 m |
| | | Visual elements | Paved ground | ≤22% |
| | | | Pedestrians and animals | ≤1% |
| | | Subjective evaluations of visual and smell environments | VECD * | ≥3.4 |
| NDSI | ≥0 | Spatial indicators | HDR * | ≥90 m |
| | | Subjective evaluations of visual and smell environments | VEDD * | ≥3.2 |
| SCD | ≥3 | Spatial indicators | VDR | ≤−10 m |
| | | | HDR | ≥70 m |
| | | Visual elements | Buildings ** | ≤13% |
| | | Subjective evaluations of visual and smell environments | SECD | ≥3.2 |
| | | | VEND | ≥3.1 |

Notes: * adjusted $R^2 < 0.3$; ** adjusted $R^2 < 0.1$.

## 5. Conclusions

This study took Chongqing as an example to investigate the current situation of soundscapes in WSMCs and discussed the influence of spatial characteristics, as well as visual and smell environments, on soundscape evaluation parameters ($L_{Aeq\_5min}$, NDSI, and SCD). The results show that the subjective and objective soundscape evaluations of WSMCs are of poor quality. Traffic sounds are dominant (33%), and natural sounds only account for 27%. Spatial indicators (elevation, VDS, VDR, and HDR) were significantly correlated with soundscape evaluation parameters. Among them, the VDR is the most influential variable on the $L_{Aeq\_5min}$ and NDSI, and the HDR is the most influential variable on the SCD. In addition, elevation and the VDS are positively correlated with the $L_{Aeq}$ and negatively correlated with the SCD. In terms of the proportions of visual elements, paved ground, pedestrians, and buildings in photos have negative effects on the soundscape, while the sky, water, and natural terrain have positive effects. Subjective evaluation results showed that high visual and smell environment quality can enhance soundscape evaluation, although the smell environment had a greater impact on SCD than the visual environment. In general, the $L_{Aeq}$ and NDSI are more affected by spatial characteristics, and the SCD is more affected by visual and smell environments in WSMCs. Finally, this study summarizes the recommended values of spatial characteristics and visual and smell environment indicators to achieve a positive soundscape evaluation. Considering the likely accelerated urban construction process in the future, the results of this study can provide effective data support and references for soundscape design and landscape environment construction in WCMCs in order to improve environmental health and people's happiness. More research is needed to further optimize model performance to improve data accuracy and to discuss the impact of different population experiences in the future.

**Author Contributions:** Conceptualization, H.X. and B.Z.; methodology, B.Z., H.X., T.G., L.Q. and H.L.; validation, H.X., B.Z., T.G. and L.Q.; investigation, B.Z., H.X., H.L. and Z.Z.; writing—original draft preparation, B.Z. and H.X.; writing—review and editing, H.X., B.Z., T.G., L.Q., H.L. and Z.Z.; visualization, B.Z.; supervision, H.X.; project administration, H.X.; funding acquisition, H.X. All authors have read and agreed to the published version of the manuscript.

**Funding:** This research was funded by the National Natural Science Foundation of China (52078077) and Northwest A&F University Youth Talent Cultivation Project (Z1010122001).

**Conflicts of Interest:** The authors declare no conflict of interest.

## Appendix A

**Table A1.** The details of the questionnaire.

| Parts | | Question | |
|---|---|---|---|
| Soundscape | Sound sources identification | Please list sound sources you noticed (limited to 8). | Open question |
| | SCD | How would you rate the comfort of the soundscape? | From 1 = "uncomfortable" to 5 = "comfortable" |
| Visual environment | VECD | How would you rate the comfort of the visual environment? | From 1 = "uncomfortable" to 5 = "comfortable" |
| | VEND | How would you rate the natural of the visual environment? | from 1 = "artificial" to 5 = "natural" |
| | VEDD | How would you rate the diversity of the visual environment? | From 1 = "simple" to 5 = "complex" |
| Smell environment | Odor identification | Please list odors you noticed (limited to 3). | Open question |
| | SECD | How would you rate the comfort of the smell environment? | From 1 = "uncomfortable" to 5 = "comfortable" |

## Appendix B

**Table A2.** Detailed specifications of all measurement equipment and associated data processing software in the study.

| Measurement | | Equipment | Equipment Specifications | Data Processing Software | Software Specifications |
|---|---|---|---|---|---|
| Objective soundscape evaluation parameters | $L_{Aeq}$ | AWA 6228$^{+}$ Sound Level Meter (Aihua Instruments Co., Ltd., China) | IEC 61672 Class 1 Measurement Range: 20 dB–142 dB (145 dB Peak) Ref.: [57] | – | – |
| | NDSI | PCM-M10 Recorder (Sony Corporation, Japan) | Sampling frequencies: 44.1 kHz Bit rate: 32 kbps–192 kbps Recoding: binaural method Ref.: [58] | Rstudio (RStudio, Inc., Boston, USA) | Packages: tuneR, soundecology Ref.: [59] |
| Spatial indicators | Elevation, VDS, HDS, VDR, HDR | YILI X28 altimeter (Hengyi Technology Co., Ltd., China) | Barometric altimetry: ≤1 m Location accuracy: ≤2 m Ref.: [60] | – | – |
| Identification of the proportions of visual elements | | Mobile phone cameras | Camera: ≥12 megapixels Image size: 4750 × 1080 pixels | The FCN model (GUC. HPSCIL, University of Geosciences, China) | Codes: Java, C++ Accuracy: 67% for actual data Ref.: [43] |

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
