# Peer review of "The Effects of Spatial Characteristics and Visual and Smell Environments on the Soundscape of Waterfront Space in Mountainous Cities"

_forests, doi:10.3390/f14010010_

Round 1

Reviewer 1 Report

Please check and possibly correct - it's not about SPL, but rather the equivalent sound level LAeq - line 61.

In the Introduction, please explain the concept of "sanswalking" very briefly. This is of course included in Chapter 2.2, however, at the beginning of the article, it may be relevant to the reader.

It is known that noise in the case of humans is an impact of a subjective nature. Does the use of architecture students for research distort the results due to their heightened perception of the landscape? Can the subjective perception of noise with the simultaneous increased perception of the landscape affect the results of the research? I don't know if the authors will share my view, but maybe it's worth mentioning in the article? Perhaps it would be worthwhile to perform similar analyzes by randomly selected people in the future.

Shouldn't there be LAeq instead of SPL - line 194.

Please specify in the description of Fig. 4 whether the LAeq values ​​are average values ​​for the area or other values ​​(e.g. maximum in the area).

Higher values ​​of the R2 coefficient could be obtained in the case of removing points that strongly differ from others. It would be necessary to analyze whether it is possible to remove such results (for example Grubbs test). This applies, for example, to NDSI on Fig. 5 b). This is a note for future research and does not need to be changed in this article. However, it can be briefly described that such an operation would increase the value of the R2 coefficient.

Please check all diagrams of Fig. 8 a), e) and indicate for which tests the coefficient R2 is given.

Author Response

Response to Reviewer 1 Comments

Point 1: Please check and possibly correct - it's not about SPL, but rather the equivalent sound level LAeq - line 61.

Response 1: Yes, we have checked the reference and replace SPL with LAeq in L68.

Point 2: In the Introduction, please explain the concept of "sanswalking" very briefly. This is of course included in Chapter 2.2, however, at the beginning of the article, it may be relevant to the reader.

Response 2: As advised, we have added a new reference [32] and explanation of "sensewalking" in L86-87.

Point 3: It is known that noise in the case of humans is an impact of a subjective nature. Does the use of architecture students for research distort the results due to their heightened perception of the landscape? Can the subjective perception of noise with the simultaneous increased perception of the landscape affect the results of the research? I don't know if the authors will share my view, but maybe it's worth mentioning in the article? Perhaps it would be worthwhile to perform similar analyzes by randomly selected people in the future.

Response 3: Yes, we added discussion on that different populations might affect the experimental results in Section 4.3, and in follow-up studies, randomized participants might be employed to verify the generality of the conclusions of this study (L496-500).

Point 4: Shouldn't there be LAeq instead of SPL - line 194.

Response 4: Yes, we have replaced SPL with LAeq in L239 (original L194).

Point 5: Please specify in the description of Fig. 4 whether the LAeq values are average values for the area or other values (e.g. maximum in the area).

Response 5: LAeq values are average values for the area or other values. We have specified it in L264-265.

Point 6: Higher values of the R2 coefficient could be obtained in the case of removing points that strongly differ from others. It would be necessary to analyze whether it is possible to remove such results (for example Grubbs test). This applies, for example, to NDSI on Fig. 5 b). This is a note for future research and does not need to be changed in this article. However, it can be briefly described that such an operation would increase the value of the R2 coefficient.

Response 6: Yes, we have added the discussion about the optimization of model in Section 4.3 (L489-496) and the outlook for future research in Section 5 (L522-524).

Point 7: Please check all diagrams of Fig. 8 a), e) and indicate for which tests the coefficient R2 is given.

Response 7: We have checked all diagrams from Fig. 8 a) to Fig. e). And coefficient R2 is given in each diagram. For Fig. 8 a) and e), the adjusted R2 is the total value of the variables in each model.

Reviewer 2 Report

Reviewer's comments:

The main objective of your paper is to investigate and explore the influence the impact of spatial characteristics and visual and olfactory environment on the soundscape perception of waterfront space in mountainous cities. I find the topic interesting, the approach is quite well done, and the manuscript is well written. After carefully reading your manuscript, I determined that your work has the potential to be accepted after a major revision. My comments are summarized in the following points:

  • Since you used sense walk as a method of your research, I invite you to add the term "Perception" to the title so that it is: “The effects of spatial characteristics and the visual and smell environment on the soundscape perception of waterfront space in mountainous cities.”

  • Then, I suggest you change every part of your manuscript that contains “the effects of spatial characteristics and the visual and smell environment on the soundscape “ to “the effects of spatial characteristics and the visual and smell environment on the soundscape perception “

  • Remove the expression "good" that appears in lines 16 and 22 and in lines 215, 251, 279, 324, 443, and 451.

  • For the keywords, I suggest not using the ones declared in the title.

  • Lines 125-127: Please replace "artificial sound" with "human sound." Additionally, I advise you to put (Biological and Geophysical sounds) in parenthesis for the natural sound category. In addition, I recommend changing the term "other sounds" to (Mechanical sounds). I also ask you to revise any part of your manuscript that contains these expressions after that. And to ensure that you agree with my comment, I ask you to consult the following papers: (Soundscape effects on visiting experience in city park: A case study in Fuzhou, China) and (Evaluation of the soundscapes through the cafe terraces before and after the COVID-19 lockdown in coastal cities in Algeria.).

  • Lines 138-140: Please specify in the manuscript if you recorded the sound using stereo, binaural, or another technique.

  • Finally, I think that your manuscript is very interesting, but your bibliographic list must contain some recent references related to “Multisensory Interaction”, “Promenade Experience” and “sense walk”.

Author Response

Response to Reviewer 2 Comments

Point 1: Since you used sense walk as a method of your research, I invite you to add the term "Perception" to the title so that it is: “The effects of spatial characteristics and the visual and smell environment on the soundscape perception of waterfront space in mountainous cities.”

Response 1: Thank you for the valuable comments! However, the soundscape is defined in ISO 129131:2014 as ‘acoustic environment as perceived or experienced and/or understood by a person or people, in context’, which contains the meaning of perception. Considering that soundscape perception might cause semantic duplication, we think it might be more appropriate to keep the original title. Thank you!

Point 2: Then, I suggest you change every part of your manuscript that contains “the effects of spatial characteristics and the visual and smell environment on the soundscape” to “the effects of spatial characteristics and the visual and smell environment on the soundscape perception”

Response 2: As stated in Response1. We think “soundscape perception” might cause semantic duplication, so it might be more appropriate to keep the original title.

Point 3: Remove the expression "good" that appears in lines 16 and 22 and in lines 215, 251, 279, 324, 443, and 451.

Response 3: Yes, we have removed the expression "good" " and replaced it with the appropriate word in L16, 22, 261, 303, 331, 506 and 514.

Point 4: For the keywords, I suggest not using the ones declared in the title.

Response 4: Yes, thank you for your kind suggestion. We have modified the keywords in L28-29.

Point 5: Lines 125-127: Please replace "artificial sound" with "human sound." Additionally, I advise you to put (Biological and Geophysical sounds) in parenthesis for the natural sound category. In addition, I recommend changing the term "other sounds" to (Mechanical sounds). I also ask you to revise any part of your manuscript that contains these expressions after that. And to ensure that you agree with my comment, I ask you to consult the following papers: (Soundscape effects on visiting experience in city park: A case study in Fuzhou, China) and (Evaluation of the soundscapes through the cafe terraces before and after the COVID-19 lockdown in coastal cities in Algeria.).

Response 5: Yes, we have modified in L148-151 and added new references [35-37].

Point 6: Lines 138-140: Please specify in the manuscript if you recorded the sound using stereo, binaural, or another technique.

Response 6: Yes. The audio was recorded by binaural method, which is specified in L178

Point 7: Finally, I think that your manuscript is very interesting, but your bibliographic list must contain some recent references related to “Multisensory Interaction”, “Promenade Experience” and “sense walk”.

Response 7: Thank you for your kind suggestion. We have added new relevant references including references [23-30], [32], [37], which related to “Multisensory Interaction”, “Promenade Experience” and “sensewalk”. 

Reviewer 3 Report

Section 1 must be improved.

-       Authors should emphasize contribution and novelty, the introduction needs to clarify the motivation, challenges, contribution, objectives, and significance/implication. 

-       You must properly introduce your work, specify well what were the goals you set yourself and how you approached the problem.

Section 2 must be improved.

-       The authors must describe in detail how the questionnaire were prepared. These are the essential elements of the work and are not adequately dealt with. Authors must specify the standards used to prepare these tests.

-       Describe in detail the equipment used to make the measurements (LAeq). Extract this data from the datasheet of the instrumentation manufacturer. To make reading the specifications of the instruments more immediate, you can insert them in a table, listing the instruments used and the specific characteristics for each.

-       Furthermore, a description of the hardware and software used for data processing is completely missing. Describe in detail the hardware used:  Extract this data from the datasheet of the hardware manufacturer. To make reading the specifications of the hardware more immediate, you can insert them in a table, listing the instruments used and the specific characteristics for each.

-       Also, you should describe in detail the software platform you used for the analysis. For example for analyze 150 visual elements.

Section 3 must be improved.

-       In section 3.1 explains in detail how you obtained the percentages of the sources, how you characterized the sound recorded near the listening points.

-       Introduce adequately the topic (Spearman’s correlation coefficients)

-       Introduce adequately the topic (multiple linear regression)

-       I could not find a detailed description of the evaluation metrics you have adopted. How will you measure your model's performance? This section is essential in order to demonstrate the effectiveness of your methodology. Furthermore, only by adopting adequate metrics will it be possible to compare your results with those obtained by other researchers.

Section 5 must be improved.

-       Paragraphs are missing where the possible practical applications of the results of this study are reported. What these results can serve the people, it is necessary to insert possible uses of this study that justify their publication.

-       They also lack the possible future goals of this work. Do the authors plan to continue their research on this topic?

42-45) Add references to support these statements.

55) Replace Watts et al. with Watts et al. [12]. I will not repeat this advice again, it also applies to the other occurrences.

69-73) Add more references to works that have already dealt with the topic, for example:” Representation of the soundscape quality in urban areas through colours”.

Author Response

Response to Reviewer 3 Comments

Section 1 must be improved.

Point 1: Section 1 must be improved. Authors should emphasize contribution and novelty, the introduction needs to clarify the motivation, challenges, contribution, objectives, and significance/implication.

Response 1: Thank you for your constructive comments! We have carefully considered the comments and make corresponding changes. In L38-43, L55-59, L73-76, and L86-92, we rewrote the statements about the motivations, contributions and novelty of the paper. In addition, we rewrote the last paragraph of introduction. In L95-102 we add the objectives, and introduction of the challenge of the study, respectively. At the end of the paragraph (L102-105) the importance and potential applications of this study have been added, too.

Point 2: Section 1 must be improved. You must properly introduce your work, specify well what were the goals you set yourself and how you approached the problem.

Response 2: As suggested, we added the aim of this study in L95-100, and introduced how we approached problems (L93-95) and faced challenges (L100-102).

Section 2 must be improved.

Point 3: The authors must describe in detail how the questionnaire were prepared. These are the essential elements of the work and are not adequately dealt with. Authors must specify the standards used to prepare these tests.

Response 3: As advised, in Section 2.2, we have added the description of the questionnaire design (L133-169) and a table introducing the details of the questionnaire (Appendix A, Table A, L534). All participants underwent a pre-investigation (L133-137) before performing the formal sensewalk. This study refers to the existing research [35-40] and feedback of pre-investigation design questionnaire. The KMO index and Cronbach's Alpha were used to test the validity and reliability (L168-169). The references are as follows:

35. Standardization, I.O.f. ISO/TS 12913-2: 2018 acoustics—soundscape—part 2: data collection and reporting requirements. ISO, Geneva 2018.

36. Berkouk, D.; Bouzir, T.A.K.; Boucherit, S.; Khelil, S.; Mahaya, C.; Matallah, M.E. Evaluation of the soundscapes through the cafe terraces before and after the COVID-19 lockdown in coastal cities in Algeria. Noise & Vibration Worldwide 2022, 53, 377-389.

37. Liu, J.; Xiong, Y.; Wang, Y.; Luo, T. Soundscape effects on visiting experience in city park: A case study in Fuzhou, China. Urban Forestry & Urban Greening 2018, 31, 38-47.

38. Russell, J.A.; Lanius, U.F. Adaptation level and the affective appraisal of environments. Journal of Environmental Psychology 1984, 4, 119-135.

39. Xiao, J. Smell, smellscape, and place-making: a review of approaches to study smellscape. Handbook of research on perception-driven approaches to urban assessment and design 2018, 240-258.

40. Henshaw, V. Urban smellscapes: Understanding and designing city smell environments; Routledge: 2013.

Point 4: Describe in detail the equipment used to make the measurements (LAeq). Extract this data from the datasheet of the instrumentation manufacturer. To make reading the specifications of the instruments more immediate, you can insert them in a table, listing the instruments used and the specific characteristics for each.

Response 4: As advised, we add Appendix B, Table B (L186, 193, 199, 205, 536), listing the equipment and associated data processing software used and the specific characteristics for each, and refer to the datasheet [43, 57-59] of the instrumentation manufacturer.

Point 5: Furthermore, a description of the hardware and software used for data processing is completely missing. Describe in detail the hardware used: Extract this data from the datasheet of the hardware manufacturer. To make reading the specifications of the hardware more immediate, you can insert them in a table, listing the instruments used and the specific characteristics for each.

Response 5: Yes. Because of the data processing software is associated with the measurement equipment in this study. Therefore, as mentioned above, we have added Appendix B, Table B (L186, 193, 199, 205, 536), listing the data processing software used and the specific characteristics. In addition, we have added a description of the minimum-required hardware for data analysis (L231-233) in Section 2.6.

Point 6: Also, you should describe in detail the software platform you used for the analysis. For example for analyze 150 visual elements.

Response 6: Yes. As mentioned in Response 4, we have added the details of the visual elements analysis software (The FCN model) to Appendix B, Table B (L563), as well as its developer and reference [43] (L200-202).

Point 7: Section 3 must be improved. In section 3.1 explains in detail how you obtained the percentages of the sources, how you characterized the sound recorded near the listening points.

Response 7: In Section 3.1 we added details on how to obtain the percentages of the sources (L236-238). And how to classify sound sources is described in Section 2.2 questionnaire design (L146-151). The reference are [35-37].

Point 8: Introduce adequately the topic (Spearman’s correlation coefficients)

Response 8: We added the discussion of Spearman's correlation coefficients (L278-282, 369-370). In addition, we also rewrote the introduction of Spearman's correlation in the methods (L212-219).

Point 9: Introduce adequately the topic (multiple linear regression) √

Response 9: As advised, we have modified L287 and added a description of evaluation metrics according to the references [26, 27, 48] (L291-294, 391-397) to properly introduce multiple linear regression. In addition, we rewrote multiple linear regression in the method (L220-229).

Point 10: I could not find a detailed description of the evaluation metrics you have adopted. How will you measure your model's performance? This section is essential in order to demonstrate the effectiveness of your methodology. Furthermore, only by adopting adequate metrics will it be possible to compare your results with those obtained by other researchers.

Response 10: As mentioned in Response 9, we added the description of evaluation metrics according to the references [26, 27, 48] (L291-294, 391-397). In addition, we added a discussion of the low performance model and the limitation of this in the discussion Section 4.3 (L489-496). Due to limited time and energy, we will discuss how to improve the performance of the model in future work (L494-496). The references to evaluation metrics are as follows:

26. Liu, J.; Kang, J.; Behm, H.; Luo, T. Effects of landscape on soundscape perception: Soundwalks in city parks. Landscape and urban planning 2014, 123, 30-40.

27. Shao, Y.; Hao, Y.; Yin, Y.; Meng, Y.; Xue, Z. Improving soundscape comfort in urban green spaces based on aural-visual interaction attributes of landscape experience. Forests 2022, 13, 1262.

48. Yang, M. A review of regression analysis methods: Establishing the quantitative relationships between subjective soundscape assessment and multiple factors; Universitätsbibliothek der RWTH Aachen: 2019.

Section 5 must be improved.

Point 11: Paragraphs are missing where the possible practical applications of the results of this study are reported. What these results can serve the people, it is necessary to insert possible uses of this study that justify their publication.

Response 11: As suggested, in Section 5 we have added the possibility of practical applications and what the results can serve people in L519-522.

Point 12: They also lack the possible future goals of this work. Do the authors plan to continue their research on this topic?

Response 12: In Section 5 we have added the future goals of this work in L522-524, which also have been discussed in L492-500.

Point 13: 42-45) Add references to support these statements.

Response 13: Yes, we have added new references [5] to support the statements in L44-47.

Point 14: 55) Replace Watts et al. with Watts et al. [12]. I will not repeat this advice again, it also applies to the other occurrences.

Response 14: Yes, We have made changes to the author names that appear in the full text according to the guidelines on the author-[reference number] nexus, and appreciate your comments.

Point 15: 69-73) Add more references to works that have already dealt with the topic, for example:” Representation of the soundscape quality in urban areas through colours”.

Response 15: Yes, thank you for pointing out the lack of references on this topic. References [26-29] (L80-82) have now been added to support this topic in this section.

Round 2

Reviewer 2 Report

Thank you for considering our comments.

Reviewer 3 Report

The authors addressed the reviewer's comments with attention and modified the paper with the suggestions provided. The new version of the paper has improved significantly both in the presentation and in the contents.